# COVID-19 in patients with hepatobiliary and pancreatic diseases: a single-centre cross-sectional study in East London

Abu Z M Dayem Ullah ![ORCID] ,[1,2] Lavanya Sivapalan,[1] Hemant M Kocher ![ORCID] ,[2,3] Claude Chelala[1]

[1]Centre for Cancer Biomarker and Biotherapeutics, Barts Cancer Institute, London, UK
[2]Barts and the London HPB Centre, The Royal London Hospital, Barts Health NHS Trust, London, UK
[3]Centre for Tumour Biology, Barts Cancer Institute, London, UK

**Correspondence to**
Dr Abu Z M Dayem Ullah;
d.ullah@qmul.ac.uk

## ABSTRACT

**Objective** To explore risk factors associated with COVID-19 susceptibility and survival in patients with pre-existing hepato–pancreato–biliary (HPB) conditions.

**Design** Cross-sectional study.

**Setting** East London Pancreatic Cancer Epidemiology (EL-PaC-Epidem) Study at Barts Health National Health Service Trust, UK. Linked electronic health records were interrogated on a cohort of participants (age ≥18 years), reported with HPB conditions between 1 April 2008 and 6 March 2020.

**Participants** EL-PaC-Epidem Study participants, alive on 12 February 2020, and living in East London within the previous 6 months (n=15 440). The cohort represents a multi-ethnic population with 51.7% belonging to the non-White background.

**Main outcome measure** COVID-19 incidence and mortality.

**Results** Some 226 (1.5%) participants had confirmed COVID-19 diagnosis between 12 February and 12 June 2020, with increased odds for men (OR 1.56; 95% CI 1.2 to 2.04) and Black ethnicity (2.04; 1.39 to 2.95) as well as patients with moderate to severe liver disease (2.2; 1.35 to 3.59). Each additional comorbidity increased the odds of infection by 62%. Substance misusers were at more risk of infection, so were patients on vitamin D treatment. The higher ORs in patients with chronic pancreatic or mild liver conditions, age >70, and a history of smoking or obesity were due to coexisting comorbidities. Increased odds of death were observed for men (3.54; 1.68 to 7.85) and Black ethnicity (3.77; 1.38 to 10.7). Patients having respiratory complications from COVID-19 without a history of chronic respiratory disease also had higher odds of death (5.77; 1.75 to 19).

**Conclusions** In this large population-based study of patients with HPB conditions, men, Black ethnicity, pre-existing moderate to severe liver conditions, six common medical multimorbidities, substance misuse and a history of vitamin D treatment independently posed higher odds of acquiring COVID-19 compared with their respective counterparts. The odds of death were significantly high for men and Black people.

## INTRODUCTION

COVID-19 is a novel infectious disease caused by SARS-CoV-2, with a wide-ranging disease

### Strengths and limitations of this study

► First multi-ethnic population-based study on COVID-19 in patients with hepato–pancreato–biliary group of diseases.

► Systematic identification of the effect, or the lack of it, of individual demographic and clinical factors on the infection and mortality of COVID-19 in a large cohort of over 15 000 patients, robustly controlling for potential confounders in their evaluation.

► Access to longitudinal data from linked primary and secondary care electronic health records, and use of rule-based phenotyping algorithms allowed for improved completeness and accuracy of the explored variables.

► Some observed increased odds of SARS-CoV-2 infection and related death could be plausibly explained by unmeasured confounding.

► The effects reported in the study could be influenced by the relatively smaller size of COVID-19 cases within this cohort.

course. Infection and mortality rates of the COVID-19 pandemic have varied widely among nations and demographics,[1] while risks are still being explored, identified and categorised according to the severity.[2 3] There are several confirmed risk factors of COVID-19 and severe outcomes, including old age,[2 4 5] chronic pulmonary disease,[2–6] cardiovascular disease,[2 5 6] hypertension,[5] chronic kidney disease,[2 4 6] diabetes mellitus,[2 5] obesity,[2 6 7] haematological diseases,[2 4] malignancy[2 4–6 8] and immunocompromised state such as HIV infection.[2 4 9] Medical complications following hospitalisation, including acute episodes of cardiovascular, respiratory, neurological, renal or hepatic failure, have also been linked to severe outcomes.[10] There are also other risk factors reported, such as smoking[11 12] or being from a Black, Asian and minority ethnic (BAME) group,[13–15] the effects of which are either mixed or the reasoning is not clearly

understood.[4] Concerns have also been raised regarding the use of various medications with respect to the risk or protective effect to COVID-19.[16–18]

Patients with diseases of the liver, pancreas or biliary tract (hepato–pancreato–biliary; HPB) are considered, in general, to be at risk of developing serious medical conditions. Expression of the ACE2 gene—a receptor for the SARS-CoV-2 virus—along the gastrointestinal tract is well documented, which suggests the digestive system is a potential route for COVID-19,[18] making patients with a diseased HPB system susceptible to this novel infection. The prevalence of COVID-19 among patients with hepatic conditions has been explored,[6 15 19] indicating severe liver disease as a moderate risk factor for COVID-19.[2] In contrast, very limited data are available on the prevalence of COVID-19 among patients with pancreatic or biliary conditions,[20] although pancreatic manifestations of the disease are rare.[21 22] It is important that clinical characteristics of COVID-19 are investigated for the HPB group as a whole, not only because these diseases demonstrate similar clinical–biological behaviours,[23] but also since they are commonly seen by a single clinical unit with specialist expertise in the management of these diseases.

The UK has been the worst affected country in Europe by COVID-19, with a reported death toll of 44 819 as of 30 June 2020.[24] At the same time, London had the highest incidence and mortality rates, with 33 775 confirmed cases and 8438 deaths.[25 26] Barts Health National Health Service (NHS) Trust (BHNT) is the largest NHS Trust in England and acts as provider of district general hospital facilities for around 2.5 million population of East London as well as a range of tertiary care services.[27] Between 1 March and 30 June, the three boroughs in East London—Tower Hamlets, Waltham Forest and Newham—had a combined age-standardised COVID-19-related mortality rate of 195 per 100 000 people. This was significantly higher than the rest of London where the age-standardised COVID-19-related mortality rate was 156 per 100 000 people.[25] East London is also one of the most ethnically diverse local areas in the country where an estimated 57% residents belong to a BAME group.[28] Significant health inequalities exist within the local population including higher rates of cancer, diabetes and obesity,[29] compared with the wider population. These conditions are not only known to be a precursor or consequence to HPB diseases, but also linked to COVID-19 and severe outcomes. In this study, we integrated primary, secondary and tertiary electronic healthcare records (EHRs) of patients with HPB diseases in East London. We inspected the demographics, lifestyle, comorbidities and associated medication use of these patients, and any possible links with SARS-CoV-2 infection. We also evaluated whether the effect of these prevalent factors as well as clinical observations during COVID-19-related hospitalisation are associated with mortality. This study will inform the management of this specific cohort of patients.

## METHODS

### Study setting and data sources

All data used for this study were collected and processed under the East London Pancreatic Cancer Epidemiology (EL-PaC-Epidem) Study at BHNT. In brief, EL-PaC-Epidem is an ongoing study that ascertains patients diagnosed or reported with HPB diseases including cancers, as well as control patients (eg, small intestine, hernia), within five BHNT hospital sites (The Royal London Hospital, Newham University Hospital, St Bartholomew's Hospital, Whipps Cross University Hospital, Mile End Hospital) between 2008 and 2021. The study is limited to the secondary use of a specified subset of patients' retrospective EHR generated during the course of normal care of these patients. It links EHRs from different data sources (via UK unique individual NHS numbers), including primary care through general practitioners (GPs) (Discovery East London Programme data service (DDS)) and secondary or tertiary care through hospitals (BHNT Consolidated Data Extract (CDE)). Patients who have previously informed their GPs or NHS to stop sharing their personal and health records for purposes other than their individual care were automatically excluded. The current EL-PaC-Epidem Study cohort consists of 27 321 adult patients (aged 18 years or over), diagnosed or reported with at least one of the HPB conditions (online supplemental table 1) between 1 April 2008 and 6 March 2020.

### Study design and population

This is a single-centre cross-sectional study using the linked EHR data of patients with a history of HPB diseases. Within this specific patient group, the study focused on the incidence of COVID-19, and examined the association of SARS-CoV-2 infection with six common medical comorbidities (ie, diabetes, hypertension, high cholesterol, cardiovascular disease, chronic respiratory disease, renal disease), lifestyle factors (ie, smoking, alcohol use, substance misuse, obesity), and use of selected prescription medications.

As the first case of COVID-19 in London was reported on 12 February 2020, we used this as the start date for this study and extracted data on a subgroup of the EL-PaC-Epidem Study cohort until 12 June 2020 (figure 1). Eligible individuals were a resident in East London and alive on the study start date (EL-HPB). Residency of East London was inferred if a patient had at least one appointment or prescription issued from a GP in East London boroughs or had a scheduled or unscheduled visit to one of the BHNT hospitals within the last 6 months (after 12 August 2019). Patients with *confirmed* SARS-CoV-2 infection were identified by: (1) the presence of International Classification of Diseases 10th edition (ICD-10) or Systematized Nomenclature of Medicine Clinical Terms (SNOMED CT) codes for confirmed COVID-19 diagnosis assigned in their hospital encounters or GP records during the observation period between 12 February and 12 June 2020 (online supplemental table 2) OR (2) positive record of

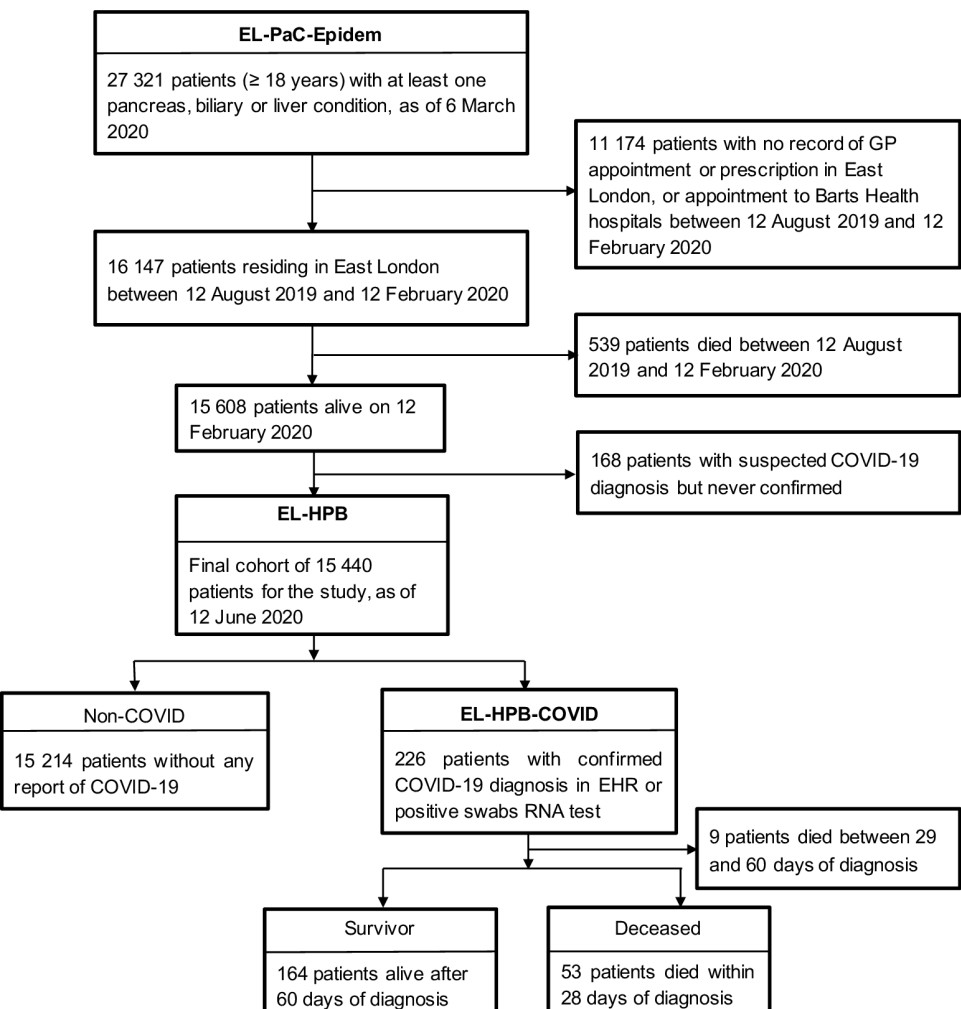

**Figure 1** Selection of patients for the cross-sectional study. EHR, electronic health record; EL-PaC-Epidem, East London Pancreatic Cancer Epidemiology; GP, general practitioner; HPB, hepato–pancreato–biliary.

SARS-CoV-2 RNA through BHNT oral and/or nasal swabs test during the same period. For confirmed COVID-19 cases, the earliest date of diagnosis or positive swab test was considered as the *index date,* whereas 12 February 2020 was considered as *index date* for rest of the cohort. Patients who were assigned an ICD-10 or SNOMED CT diagnosis code for *suspected* COVID-19, but were neither reassigned to confirmed diagnosis nor positive RNA test, were excluded from the analysis.

We also examined the onset-to-death distribution within the patient group with a confirmed COVID-19 diagnosis (EL-HPB-COVID). Mortality data were collected on 12 October 2020. Following the latest Public Health England definition,[30] the death of a patient within 28 days of the index date is considered as a COVID-19-related death. This is different from a 60-day window that was being used in the UK prior to 12 August 2020 to define COVID-19-related death. To ensure consistency, patients with COVID-19 who survived beyond 60 days of index date are considered as survivors in the study; nine patients who died between 29 and 60 days of diagnosis were excluded from the analysis. The onset-to-death distribution was analysed in the context of the same set of comorbidities,

lifestyle factors and medication use, as well as cardiovascular, respiratory and renal complications during hospital care.

## Procedures

All patient data were obtained from retrospective EHR, harmonised across hospital and GP coding systems where applicable, and organised into 40 primary variables across seven categories corresponding to the focus of the study (table 1). BHNT CDE uses 2011 UK census grouping to record ethnicity, ICD-10 or SNOMED diagnosis codes for clinically relevant diagnoses, and Office of the Population Censuses and Surveys Classification of Interventions and Procedures V.4 (OPCS-4) procedural codes for treatments and procedures. Physiological observations (weight, body mass index (BMI), blood pressure) and laboratory test results are available in locally developed terms. Semistructured text entries such as discharge summaries, medical history and a lifestyle questionnaire collected during the preoperative assessment, and presenting symptoms from scheduled or unscheduled hospital visits are also available. All GP records via DDS were available in Read Codes V.2 or Clinical Terminology V.3 (CTV3) codes,

| Table 1 | Variables and outcomes explored in this study | |
|---|---|---|
| Category | Variables | Levels/units |
| Demographic | Gender | Female, male |
| | Ethnicity | White, South Asian, Black, other, (not available) |
| | Age (continuous) | Years |
| | Age group* | 18–40, 41–50, 51–60, 61–70, 71–80, 80+ |
| | Binary age group* | 18–60, 60+ |
| HPB disease | Cancer | No, yes |
| | Pancreatic disease | Acute, chronic |
| | Biliary disease | Acute, chronic |
| | Liver disease | Mild, moderate/severe |
| Comorbidity | Diabetes | No, yes |
| | Hypertension | No, yes |
| | High cholesterol | No, yes |
| | Cardiovascular disease | No, yes |
| | Chronic respiratory disease | No, yes |
| | Renal disease | No, yes |
| | Number of comorbidities* | None, 1, 2, 3 or more |
| Lifestyle factors | Smoker | Never, past, current, (not available) |
| | Alcohol drinker | Never, past, current, (not available) |
| | Substance user | Never, past, current, (not available) |
| | Obese | Never, past, current, (not available) |
| Medication use | Angiotensin-convertingenzyme (ACE) inhibitors | Non-user, past, current user |
| | Angiotensin II receptor blocker inhibitors (ARB) | Non-user, past, current user |
| | Aldosterone antagonists (MCRA) | Non-user, past, current user |
| | Beta-adrenergic blocking agents (β-blocker) | Non-user, past, current user |
| | Calcium channel blockers (CCB) | Non-user, past, current user |
| | Alpha agonist (α-agonist) | Non-user, past, current user |
| | Thiazide | Non-user, past, current user |
| | Antiplatelet | Non-user, past, current user |
| | Antiarrhythmic | Non-user, past, current user |
| | Anticoagulant | Non-user, past, current user |
| | Glucocorticoid | Non-user, past, current user |
| | Beta-2 adrenergic receptor agonists ($β_2$-agonist) | Non-user, past, current user |
| | Muscarinic antagonist | Non-user, past, current user |
| | Non-steroidal anti-inflammatory drugs (NSAID) | Non-user, past, current user |
| | Vitamin D | Non-user, past, current user |
| | Proton pump inhibitors (PPI) | Non-user, past, current user |
| | Statin | Non-user, past, current user |
| | Immunosuppressant | Non-user, past, current user |
| Complications | Cardiovascular | No, recurrent, novel |
| | Respiratory | No, recurrent, novel |
| | Renal | No, recurrent, novel |
| | Number of recurrent complications* | None, 1, 2, 3 |
| | Number of novel complications* | None, 1, 2, 3 |
| Outcome | COVID-19 incidence | Non-COVID-19, COVID-19 |
| | COVID-19 mortality | Survivor, deceased |

Continued

**Table 1** Continued

| Category | Variables | Levels/units |
|---|---|---|

All variables are categorical, unless otherwise stated. For categorical variables, the first value represents the reference level. HPB diagnosis groups are independent binary categorical variables.
*Derived variables.
HPB, hepato–pancreato–biliary; MCRA, mineralocorticoid receptor antagonist.

except the prescribed medication records and COVID-19 diagnosis which were available in SNOMED codes. For each variable, we consulted ICD-10, SNOMED, Read, CTV3 or OPCS-4 dictionaries as appropriate to construct the mapping *codelists*. For some variables, codelists also included keywords to conduct automated substring search within semistructured text as well as local laboratory test and physiological observation terms.

Rule-based phenotyping algorithms were developed for each categorical variable to characterise patients, integrating information from multiple sources where available to counteract bias. HPB diseases were grouped into four categories (online supplemental table 1): *any* malignant disease, and non-malignant diseases of liver, pancreas or biliary tract. Non-malignant liver diseases were further divided into mild and moderate to severe subgroups, extending the definition from CDMF Charlson Comorbidity Index,[31] whereas non-malignant pancreas or biliary diseases were divided into acute and chronic disease subgroups (online supplemental table 1). Within each disease category, a patient was assigned to chronic (or more severe) subgroup, when data indicated the history of both acute (or mild) and chronic (more severe) conditions. A patient can either be assigned to a malignant disease category or any of the non-malignant disease subgroups. Ethnicity was grouped into four categories—White, South Asian, Black and Other. White and Black ethnic groups were defined based on the 2011 UK census classification; Indian, Pakistani and Bangladeshi origin from the Asian group represented South Asian, while the rest (ie, Mixed, Chinese, other Asian and other ethnic group) were represented in the Other group. The ethnic category recorded at the GP took precedence over hospital records.

Phenotyping algorithms defining the comorbidities were based on diagnosis codes (presence) or semistructured text search (presence or absence), with the additional inclusion of procedural codes (presence), some observation or laboratory test results (presence) and related medication use (at least three prescriptions). Patients were considered to have or have had a specific medical condition if they met at least one criterion indicating the presence of the condition before the *index date*, otherwise they were considered negative for the condition.

Phenotyping algorithms defining the lifestyle factors were based on the longitudinal entries (current, past or never) derived from diagnosis codes and free text search, with the additional inclusion of BMI observation for obesity. Obesity was defined as BMI of 30 kg/m² or more. Patients assigned *never* status at any point but having a record of *current* or

*past* status before that date were reassigned to *past* status. The most recent lifestyle record before or on the *index date* was then used to assign *current, past* or *never* status to the patients. Patients with no record of a specific lifestyle factor were classified as having missing data. Patients were assigned *current, past* or *non-user* status for medication use variables based on the number of GP prescriptions issued in the last 2 years for the medicines under specific medication groups. Patients with no record of prescription for particular medications were assigned *non-user* status. With at least three prescriptions issued, a patient was assigned *current user* status if the latest issue was within 3 months preceding the *index date*, and *past user* status otherwise. Patients with record of less than three prescriptions were classified as *non-user*. Patients with COVID-19 were considered to have a specific complication during admitted patient care if at least one of the hospital diagnosis codes from the complications *codelist* was recorded during the observation period after *index date*, otherwise they were considered negative for the complication. A patient was considered to have a *recurrent* complication if they had a history of that particular comorbidity, otherwise it was considered as a *novel* complication.

Selection of study variables, *codelist* construction, and phenotyping algorithm development were done in consultation with a panel of clinicians and scientists (HK, CC, LS). A comprehensive list of codelists and phenotyping algorithms for the study variables are available on the EL-PaC-Epidem portal (https://pac-epidem-el.bcc.qmul.ac.uk/covid-19/).

## Statistical analysis

We conducted descriptive analyses for the EL-HPB cohort as a whole, by a group for patients with confirmed SARS-CoV-2 infection and the rest (herein referred to as COVID-19 and non-COVID-19, respectively). Differences in demographic and clinical characteristics between the groups were assessed with Pearson's $\chi^2$ test, Fisher's exact test and Kruskal-Wallis rank-sum test, as appropriate. P values less than 0.05 were considered significant. Similar descriptive analyses were performed for the EL-HPB-COVID cohort, and by survivor and deceased groups.

To explore the risk factors associated with COVID-19 susceptibility and subsequent survival, the effect size for each variable under investigation was evaluated with ORs with 95% CIs, using regression models with a binomial distribution. Crude ORs were obtained from univariable regression models, and then simultaneously controlled for a fixed set of potential confounders (gender, ethnicity, age group) using multivariable regression models with

Benjamini-Hochberg correction for p value adjustment. The median age of the overall EL-HPB cohort being 57 years, a simplified binary age grouping (18–60, 61+ years) was used in multivariable regression models for comorbidity, lifestyle, medication use and post-diagnosis complication analyses. Since a participant with non-malignant HPB diagnoses for multiple organs can be represented in multiple HPB subgroups, the effect estimation for individual HPB disease variables was further mutually controlled for other HPB diseases.

We also conducted more in-depth post hoc analysis to evaluate the confounding effect of pre-existing medical conditions by adding comorbidity covariates individually in the multivariable regression models. Finally, effect modification by non-malignant HPB disease subgroups was evaluated by adding interaction terms in the models and comparing them with models lacking this interaction via the likelihood ratio test. Any potential association between HPB diseases and COVID-19 susceptibility/mortality risk factors was further evaluated in stratified analyses according to the disease subgroups.

Patients with missing data for individual categorical variables were included in the descriptive analyses and in regression models for effect estimation. All statistical analyses and visualisations were performed in R (V.3.5.1).

### Patient and public involvement

Patients and the public were involved in evaluating the design of the umbrella study (EL-PaC-Epidem), particularly the notion of collection and processing of retrospective patient data without their consent. The support from NHS Confidentiality Advisory Group was obtained based on the positive opinion posed by patients and the public.

### RESULTS
### Population characteristics

The final EL-HPB cohort consisted of 15 540 patients, after applying the eligibility criteria and excluding 168 suspected but unconfirmed COVID-19 cases. By 12 June 2020, 226 (1.5%; 145 per 10 000 adult population) confirmed cases of COVID-19 were reported in this cohort (figure 1). This was more than three times higher than in the general population of East London where prevalence of COVID-19 at the same time was 41 per 10 000 adult population.[25] More than half of the COVID-19 cases had some form of non-malignant liver diseases (N=138, 62.8%); however, when comparing confirmed COVID-19 cases with the non-COVID-19 cases, we observed a disproportionate infection frequency in patients with chronic pancreatic conditions (14.1% vs 8.8%) and moderate to severe liver conditions (11.4% vs 6.9%). We also observed differences in gender, ethnic origin, and age group between COVID-19 and non-COVID-19 cases (table 2). The proportion of men was higher in the COVID-19 group compared with the baseline non-COVID-19 group (53.5% vs 43.7%, p=0.005). The same trend was observed for Black population (17.7% vs 10.7%). Patients with

COVID-19 were older than patients without COVID-19 (median 67 years, IQR 55.1–80.9 years vs 57.1 years, 44.8–69.2 years, p<0.001), with a steady increase in infection frequency with age. Some 78.3% of patients with COVID-19 had three or more comorbidities, with hypertension being the most common comorbidity (85.4%), followed by high cholesterol and diabetes (table 2). Only eight patients with COVID-19 had none of the six medical conditions. In general, patients with COVID-19 had a higher rate of history of smoking, drinking, substance misuse and obesity compared with the non-COVID-19 group. Consistent with the underlying prevalent comorbidities of the COVID-19 group, a history of prescription drugs use associated with managing hypertension or cardiovascular disease (ACE inhibitor, calcium channel blocker, β-blocker, aldosterone antagonists, antiarrhythmic, antiplatelet, anticoagulant), cholesterol (statin), inflammation (glucocorticoid, β2-agonists) or background HPB condition (proton pump inhibitor) was higher in patients with COVID-19 (table 2). Intake of vitamin D was also significantly higher in patients with COVID-19.

Between 12 February and 12 August 2020, the all-cause mortality rate in the non-COVID-19 group was 2.4%, whereas the rate in the COVID-19 group during the same period was 27.4% (table 2). When analysing the 53 deceased and 164 surviving patients with confirmed SARS-CoV-2 infection, we found differences in gender (p=0.005) and age (p<0.001); deceased patients were older than the survivors (median 80.4 years, IQR 71.7–85.1 years vs 62.9 years, 49.8–77.4 years) with steady increase in death with age becoming prominent in those above 70 years of age (table 3). We observed a higher mortality among South Asian (34% vs 29.3%) and Black (26.4% vs 13.4%) populations, which were even more pronounced when comparing with the all-cause mortality in the non-COVID-19 group (online supplemental table 3). Higher mortality was observed for patients with pancreatic and biliary disease in general, whereas patients with liver disease had higher survival rate (table 3). The median survival period for the deceased patients from the date of confirmed COVID-19 diagnosis was 11 days (IQR 2–18 days). After stratifying patients according to the comorbidities investigated, the mortality for patients with HPB diseases with COVID-19 was at least six times higher than that of patients with HPB diseases without COVID-19 (online supplemental table 3). Diabetes, hypertension, cardiovascular and renal conditions, in particular, were associated with mortality in patients with COVID-19 (table 3). All except one deceased patient had at least three additional comorbidities, compared with 71.3% of patients who survived. The deceased group had higher proportion of patients with a history of smoking and current substance misuse, but no overall differences were observed for drinking and obesity. Notable differences were observed in the use of glucocorticoid, β2-agonists and statins. Recurrent complications were more common in the deceased group compared with survivors, however

**Table 2** Differences in demographic, comorbidity, lifestyle, and medication use characteristics between COVID-19 infected and non-COVID-19 groups

| | Non-COVID-19 (N=15214) | COVID-19 (N=226) | Total (N=15440) | P value |
|---|---|---|---|---|
| *Demographics* | | | | |
| Gender | | | | 0.005 |
| Female | 8570 (56.3%) | 105 (46.5%) | 8675 (56.2%) | |
| Male | 6644 (43.7%) | 121 (53.5%) | 6765 (43.8%) | |
| Ethnic origin | | | | 0.004 |
| White | 6914 (45.4%) | 97 (42.9%) | 7011 (45.4%) | |
| South Asian | 4381 (28.8%) | 68 (30.1%) | 4449 (28.8%) | |
| Black | 1635 (10.7%) | 40 (17.7%) | 1675 (10.8%) | |
| Other | 1855 (12.2%) | 19 (8.4%) | 1874 (12.1%) | |
| Unknown | 429 (2.8%) | 2 (0.9%) | 431 (2.8%) | |
| HPB cancer | | | | 1 |
| No | 14779 (97.1%) | 220 (97.3%) | 14999 (97.1%) | |
| Yes | 435 (2.9%) | 6 (2.7%) | 441 (2.9%) | |
| Pancreatic disease* | | | | 0.017 |
| No | 12264 (83.0%) | 170 (77.3%) | 12434 (82.9%) | |
| Acute | 1211 (8.2%) | 19 (8.6%) | 1230 (8.2%) | |
| Chronic | 1304 (8.8%) | 31 (14.1%) | 1335 (8.9%) | |
| Liver disease* | | | | 0.004 |
| No | 6781 (45.9%) | 82 (37.3%) | 6863 (45.8%) | |
| Mild | 6985 (47.3%) | 113 (51.4%) | 7098 (47.3%) | |
| Moderate/severe | 1013 (6.9%) | 25 (11.4%) | 1038 (6.9%) | |
| Biliary disease* | | | | 0.159 |
| No | 7589 (51.3%) | 127 (57.7%) | 7716 (51.4%) | |
| Acute | 738 (5.0%) | 11 (5.0%) | 749 (5.0%) | |
| Chronic | 6452 (43.7%) | 82 (37.3%) | 6534 (43.6%) | |
| Age | | | | <0.001 |
| Median | 57.08 | 67.03 | 57.22 | |
| Q1, Q3 | 44.76, 69.19 | 55.07, 80.93 | 44.86, 69.42 | |
| Age group | | | | <0.001 |
| 18–40 | 2803 (18.4%) | 23 (10.2%) | 2826 (18.3%) | |
| 41–50 | 2714 (17.8%) | 26 (11.5%) | 2740 (17.7%) | |
| 51–60 | 3407 (22.4%) | 35 (15.5%) | 3442 (22.3%) | |
| 61–70 | 2957 (19.4%) | 42 (18.6%) | 2999 (19.4%) | |
| 71–80 | 1980 (13.0%) | 43 (19.0%) | 2023 (13.1%) | |
| 80+ | 1353 (8.9%) | 57 (25.2%) | 1410 (9.1%) | |
| All-cause mortality | | | | <0.001 |
| Survivor | 14845 (97.6%) | 164 (72.6%) | 15009 (97.2%) | |
| Deceased | 369 (2.4%) | 62 (27.4%) | 431 (2.8%) | |
| *Comorbidities* | | | | |
| Diabetes | 5854 (38.5%) | 148 (65.5%) | 6002 (38.9%) | <0.001 |
| Hypertension | 9759 (64.1%) | 193 (85.4%) | 9952 (64.5%) | <0.001 |
| Cholesterol | 8227 (54.1%) | 156 (69.0%) | 8383 (54.3%) | <0.001 |
| Cardiovascular | 4283 (28.2%) | 131 (58.0%) | 4414 (28.6%) | <0.001 |

Continued

| Table 2 | Continued | | | |
|---|---|---|---|---|
| | **Non-COVID-19** | **COVID-19** | **Total** | |
| | **(N=15214)** | **(N=226)** | **(N=15440)** | **P value** |
| Renal | 3094 (20.3%) | 110 (48.7%) | 3204 (20.8%) | <0.001 |
| Respiratory | 4574 (30.1%) | 111 (49.1%) | 4685 (30.3%) | <0.001 |
| Number of comorbidities | | | | <0.001 |
| None | 2410 (15.8%) | 8 (3.5%) | 2418 (15.7%) | |
| 1 | 2924 (19.2%) | 13 (5.8%) | 2937 (19.0%) | |
| 2 | 3039 (20.0%) | 28 (12.4%) | 3067 (19.9%) | |
| 3 or more | 6841 (45.0%) | 177 (78.3%) | 7018 (45.5%) | |
| *Lifestyle factors* | | | | |
| Smoker | | | | <0.001 |
| Not available | 436 (2.9%) | 2 (0.9%) | 438 (2.8%) | |
| Never | 6425 (42.2%) | 84 (37.2%) | 6509 (42.2%) | |
| Past | 5110 (33.6%) | 114 (50.4%) | 5224 (33.8%) | |
| Current | 3243 (21.3%) | 26 (11.5%) | 3269 (21.2%) | |
| Drinker | | | | 0.021 |
| Not available | 2505 (16.5%) | 28 (12.4%) | 2533 (16.4%) | |
| Never | 3857 (25.4%) | 58 (25.7%) | 3915 (25.4%) | |
| Past | 2145 (14.1%) | 47 (20.8%) | 2192 (14.2%) | |
| Current | 6707 (44.1%) | 93 (41.2%) | 6800 (44.0%) | |
| Substance user | | | | <0.001 |
| Not available | 7686 (50.5%) | 99 (43.8%) | 7785 (50.4%) | |
| Never | 3606 (23.7%) | 29 (12.8%) | 3635 (23.5%) | |
| Past | 403 (2.6%) | 13 (5.8%) | 416 (2.7%) | |
| Current | 3519 (23.1%) | 85 (37.6%) | 3604 (23.3%) | |
| Obese | | | | <0.001 |
| Not available | 406 (2.7%) | 1 (0.4%) | 407 (2.6%) | |
| Never | 6715 (44.1%) | 85 (37.6%) | 6800 (44.0%) | |
| Past | 2199 (14.5%) | 51 (22.6%) | 2250 (14.6%) | |
| Current | 5894 (38.7%) | 89 (39.4%) | 5983 (38.8%) | |
| *Prescription medication use* | | | | |
| ACE inhibitor | | | | <0.001 |
| Non-user | 12024 (79.0%) | 161 (71.2%) | 12185 (78.9%) | |
| Past user | 518 (3.4%) | 28 (12.4%) | 546 (3.5%) | |
| Current user | 2672 (17.6%) | 37 (16.4%) | 2709 (17.5%) | |
| Angiotensin receptor blocker | | | | 0.024 |
| Non-user | 13530 (88.9%) | 188 (83.2%) | 13718 (88.8%) | |
| Past user | 227 (1.5%) | 5 (2.2%) | 232 (1.5%) | |
| Current user | 1457 (9.6%) | 33 (14.6%) | 1490 (9.7%) | |
| Aldosterone antagonist | | | | <0.001 |
| Non-user | 14651 (96.3%) | 205 (90.7%) | 14856 (96.2%) | |
| Past user | 137 (0.9%) | 9 (4.0%) | 146 (0.9%) | |
| Current user | 426 (2.8%) | 12 (5.3%) | 438 (2.8%) | |
| β-blocker | | | | <0.001 |
| Non-user | 12161 (79.9%) | 145 (64.2%) | 12306 (79.7%) | |
| Past user | 410 (2.7%) | 12 (5.3%) | 422 (2.7%) | |

**Table 2** Continued

| | Non-COVID-19 (N=15 214) | COVID-19 (N=226) | Total (N=15 440) | P value |
|---|---|---|---|---|
| Current user | 2643 (17.4%) | 69 (30.5%) | 2712 (17.6%) | |
| Calcium channel blocker | | | | 0.005 |
| Non-user | 11 714 (77.0%) | 158 (69.9%) | 11 872 (76.9%) | |
| Past user | 581 (3.8%) | 17 (7.5%) | 598 (3.9%) | |
| Current user | 2919 (19.2%) | 51 (22.6%) | 2970 (19.2%) | |
| α-agonist | | | | 0.837 |
| Non-user | 15 131 (99.5%) | 225 (99.6%) | 15 356 (99.5%) | |
| Past user | 23 (0.2%) | 0 (0.0%) | 23 (0.1%) | |
| Current user | 60 (0.4%) | 1 (0.4%) | 61 (0.4%) | |
| Thiazide | | | | 0.759 |
| Non-user | 15 131 (99.5%) | 225 (99.6%) | 15 356 (99.5%) | |
| Past user | 32 (0.2%) | 0 (0.0%) | 32 (0.2%) | |
| Current user | 51 (0.3%) | 1 (0.4%) | 52 (0.3%) | |
| Antiplatelet | | | | <0.001 |
| Non-user | 12 512 (82.2%) | 147 (65.0%) | 12 659 (82.0%) | |
| Past user | 446 (2.9%) | 10 (4.4%) | 456 (3.0%) | |
| Current user | 2256 (14.8%) | 69 (30.5%) | 2325 (15.1%) | |
| Antiarrhythmic | | | | <0.001 |
| Non-user | 14 440 (94.9%) | 199 (88.1%) | 14 639 (94.8%) | |
| Past user | 156 (1.0%) | 7 (3.1%) | 163 (1.1%) | |
| Current user | 618 (4.1%) | 20 (8.8%) | 638 (4.1%) | |
| Anticoagulant | | | | 0.008 |
| Non-user | 14 613 (96.0%) | 208 (92.0%) | 14 821 (96.0%) | |
| Past user | 144 (0.9%) | 5 (2.2%) | 149 (1.0%) | |
| Current user | 457 (3.0%) | 13 (5.8%) | 470 (3.0%) | |
| Glucocorticoid | | | | <0.001 |
| Non-user | 10 878 (71.5%) | 122 (54.0%) | 11 000 (71.2%) | |
| Past user | 1278 (8.4%) | 22 (9.7%) | 1300 (8.4%) | |
| Current user | 3058 (20.1%) | 82 (36.3%) | 3140 (20.3%) | |
| β2-agonist | | | | <0.001 |
| Non-user | 13 443 (88.4%) | 172 (76.1%) | 13 615 (88.2%) | |
| Past user | 286 (1.9%) | 8 (3.5%) | 294 (1.9%) | |
| Current user | 1485 (9.8%) | 46 (20.4%) | 1531 (9.9%) | |
| Muscarinic antagonist | | | | <0.001 |
| Non-user | 13 531 (88.9%) | 175 (77.4%) | 13 706 (88.8%) | |
| Past user | 300 (2.0%) | 10 (4.4%) | 310 (2.0%) | |
| Current user | 1383 (9.1%) | 41 (18.1%) | 1424 (9.2%) | |
| NSAID | | | | 0.117 |
| Non-user | 13 703 (90.1%) | 198 (87.6%) | 13 901 (90.0%) | |
| Past user | 756 (5.0%) | 10 (4.4%) | 766 (5.0%) | |
| Current user | 755 (5.0%) | 18 (8.0%) | 773 (5.0%) | |
| Vitamin D | | | | <0.001 |
| Non-user | 12 542 (82.4%) | 139 (61.5%) | 12 681 (82.1%) | |
| Past user | 573 (3.8%) | 18 (8.0%) | 591 (3.8%) | |

Continued

**Table 2** Continued

| | Non-COVID-19 | COVID-19 | Total | |
|---|---|---|---|---|
| | (N=15214) | (N=226) | (N=15440) | P value |
| Current user | 2099 (13.8%) | 69 (30.5%) | 2168 (14.0%) | |
| Proton pump inhibitor | | | | <0.001 |
| Non-user | 8332 (54.8%) | 85 (37.6%) | 8417 (54.5%) | |
| Past user | 1167 (7.7%) | 15 (6.6%) | 1182 (7.7%) | |
| Current user | 5715 (37.6%) | 126 (55.8%) | 5841 (37.8%) | |
| Statin | | | | <0.001 |
| Non-user | 9128 (60.0%) | 86 (38.1%) | 9214 (59.7%) | |
| Past user | 592 (3.9%) | 16 (7.1%) | 608 (3.9%) | |
| Current user | 5494 (36.1%) | 124 (54.9%) | 5618 (36.4%) | |
| Immunosuppressant | | | | 0.103 |
| Non-user | 14722 (96.8%) | 213 (94.2%) | 14935 (96.7%) | |
| Past user | 204 (1.3%) | 5 (2.2%) | 209 (1.4%) | |
| Current user | 288 (1.9%) | 8 (3.5%) | 296 (1.9%) | |

Values are n (%), unless otherwise specified.
*Patients with HPB cancer were not included in the non-malignant disease groups.
HPB, hepato–pancreato–biliary; NSAID, non-steroidal anti-inflammatory drug.

frequency of novel respiratory complications was notably higher in the deceased group (39.6% vs 21.3%).

### Odds of SARS-CoV-2 infection

The risk analyses showed a greater odds of COVID-19 for men, the Black community and those with moderate to severe liver disease (figure 2). Patients with chronic pancreatic and mild liver conditions were also associated with a higher odds of infection (OR 1.89, 95% CI 1.25 to 2.85, p=0.007; 1.52, 1.07 to 2.15, p=0.039); however, post-hoc adjustment for comorbidities returned a reduced non-significant positive odds (1.57, 1.04 to 2.38, p=0.084; 1.32, 0.93 to 1.88, p=0.237), with diabetes principally responsible for this reduction (online supplemental table 4). The similar association was observed for elderly patients (over 70 years) with underlying multimorbidity as confounding factor. Patients with pre-existing renal conditions had the highest odds of COVID-19 (2.93, 2.2 to 3.89, p<0.001), followed by more than twofold increased odds for patients with hypertension, diabetes, cardiovascular or chronic respiratory disease (figure 2). However, the independent effects of hypertension and high cholesterol were absent when controlled for other comorbidities (online supplemental table 4).

Substance misusers had higher odds of infection, but the higher odds observed for those with a history of smoking or obesity were due to underlying comorbidities. Patients on vitamin D treatment and past users of ACE inhibitors were associated with higher odds of infection. The slightly reduced yet significantly high odds remained after controlling for comorbidities, with renal (for vitamin D users) and cardiovascular (for ACE inhibitor users) diseases being the principal source of the

reduced estimates (online supplemental table 4). Higher odds were also observed for users of PPIs, glucocorticoid, β-blockers, β2-agonists, aldosterone antagonist, muscarinic antagonist, antiplatelet, antiarrhythmic and statin compared with the non-users of these respective drugs; however, post-hoc adjustment for comorbidities returned non-significant positive odds for those.

A small number of factors appeared to modify the association between HPB disease subgroups and risk of COVID-19 infection (online supplemental table 5). In patients with mild liver disease, the odds of COVID-19 infection doubled for patients with chronic pancreatic disease compared with patients with no pancreatic condition (p value for heterogeneity, p-het by liver disease=0.02). A history of substance misuse was associated with significantly higher odds of infection, particularly for patients with chronic biliary conditions (p-het by biliary disease=0.03), and mild liver conditions (p-het by liver disease=0.04).

### Odds of COVID-19-related death

The risk analyses showed an increased odds of COVID-19-related death for men, individuals from the Black community and patients who had acute respiratory complications during admitted care without a history of long-standing respiratory problems (figure 3). Increased odds of death were also observed for the glucocorticoid and β2-agonists. No HPB disease subgroups were particularly more vulnerable to COVID-19-related death, although patients with chronic pancreatic condition showed a trend towards significance. Elderly patients (over 70 years), and recent users of ACE inhibitors and non-steroidal anti-inflammatory drugs were associated

**Table 3** Differences in demographic, comorbidity, lifestyle, medication use, and post-diagnosis complication characteristics between COVID-19 survivor and deceased groups

| | Survivor (N=164) | Deceased (N=53) | Total (N=217) | P value |
|---|---|---|---|---|
| *Demographics* | | | | |
| Gender | | | | 0.005 |
| Female | 82 (50.0%) | 15 (28.3%) | 97 (44.7%) | |
| Male | 82 (50.0%) | 38 (71.7%) | 120 (55.3%) | |
| Ethnic origin | | | | 0.053 |
| White | 74 (45.1%) | 20 (37.7%) | 94 (43.3%) | |
| South Asian | 48 (29.3%) | 18 (34.0%) | 66 (30.4%) | |
| Black | 22 (13.4%) | 14 (26.4%) | 36 (16.6%) | |
| Other | 18 (11.0%) | 1 (1.9%) | 19 (8.8%) | |
| Unknown | 2 (1.2%) | 0 (0.0%) | 2 (0.9%) | |
| Age | | | | <0.001 |
| Median | 62.94 | 80.38 | 67.17 | |
| Q1, Q3 | 49.81, 77.38 | 71.72, 85.12 | 55.00, 81.07 | |
| Age group | | | | <0.001 |
| 18–40 | 21 (12.8%) | 1 (1.9%) | 22 (10.1%) | |
| 41–50 | 23 (14.0%) | 2 (3.8%) | 25 (11.5%) | |
| 51–60 | 31 (18.9%) | 2 (3.8%) | 33 (15.2%) | |
| 61–70 | 33 (20.1%) | 8 (15.1%) | 41 (18.9%) | |
| 71–80 | 25 (15.2%) | 16 (30.2%) | 41 (18.9%) | |
| 80+ | 31 (18.9%) | 24 (45.3%) | 55 (25.3%) | |
| HPB cancer | | | | 0.594 |
| No | 161 (98.2%) | 51 (96.2%) | 212 (97.7%) | |
| Yes | 3 (1.8%) | 2 (3.8%) | 5 (2.3%) | |
| Pancreatic disease* | | | | 0.039 |
| No | 129 (80.1%) | 33 (64.7%) | 162 (76.4%) | |
| Acute | 14 (8.7%) | 5 (9.8%) | 19 (9.0%) | |
| Chronic | 18 (11.2%) | 13 (25.5%) | 31 (14.6%) | |
| Liver disease* | | | | 0.039 |
| No | 54 (33.5%) | 27 (52.9%) | 81 (38.2%) | |
| Mild | 88 (54.7%) | 19 (37.3%) | 107 (50.5%) | |
| Moderate/severe | 19 (11.8%) | 5 (9.8%) | 24 (11.3%) | |
| Biliary disease* | | | | 0.101 |
| No | 99 (61.5%) | 26 (51.0%) | 125 (59.0%) | |
| Acute | 5 (3.1%) | 5 (9.8%) | 10 (4.7%) | |
| Chronic | 57 (35.4%) | 20 (39.2%) | 77 (36.3%) | |
| Survival | | | | <0.001 |
| Median | 47 | 11 | 35 | |
| Q1, Q3 | 28.75, 66.00 | 2.00, 18.00 | 13.00, 59.00 | |
| *Comorbidities* | | | | |
| Diabetes | 99 (60.4%) | 42 (79.2%) | 141 (65.0%) | 0.012 |
| Hypertension | 132 (80.5%) | 52 (98.1%) | 184 (84.8%) | 0.002 |
| Cholesterol | 106 (64.6%) | 41 (77.4%) | 147 (67.7%) | 0.085 |
| Cardiovascular | 84 (51.2%) | 43 (81.1%) | 127 (58.5%) | <0.001 |

Continued

**Table 3** Continued

| | Survivor (N=164) | Deceased (N=53) | Total (N=217) | P value |
|---|---|---|---|---|
| Renal | 67 (40.9%) | 38 (71.7%) | 105 (48.4%) | <0.001 |
| Respiratory | 80 (48.8%) | 27 (50.9%) | 107 (49.3%) | 0.784 |
| Number of comorbidities | | | | <0.001 |
| None | 8 (4.9%) | 0 (0.0%) | 8 (3.7%) | |
| 1 | 13 (7.9%) | 0 (0.0%) | 13 (6.0%) | |
| 2 | 26 (15.9%) | 1 (1.9%) | 27 (12.4%) | |
| 3 or more | 117 (71.3%) | 52 (98.1%) | 169 (77.9%) | |
| *Lifestyle factors* | | | | |
| Smoker | | | | 0.008 |
| Not available | 2 (1.2%) | 0 (0.0%) | 2 (0.9%) | |
| Never | 67 (40.9%) | 14 (26.4%) | 81 (37.3%) | |
| Past | 72 (43.9%) | 37 (69.8%) | 109 (50.2%) | |
| Current | 23 (14.0%) | 2 (3.8%) | 25 (11.5%) | |
| Drinker | | | | 0.897 |
| Not available | 21 (12.8%) | 6 (11.3%) | 27 (12.4%) | |
| Never | 44 (26.8%) | 12 (22.6%) | 56 (25.8%) | |
| Past | 32 (19.5%) | 12 (22.6%) | 44 (20.3%) | |
| Current | 67 (40.9%) | 23 (43.4%) | 90 (41.5%) | |
| Substance user | | | | 0.055 |
| Not available | 77 (47.0%) | 19 (35.8%) | 96 (44.2%) | |
| Never | 24 (14.6%) | 4 (7.5%) | 28 (12.9%) | |
| Past | 10 (6.1%) | 2 (3.8%) | 12 (5.5%) | |
| Current | 53 (32.3%) | 28 (52.8%) | 81 (37.3%) | |
| Obese | | | | 0.184 |
| Not available | 0 (0.0%) | 1 (1.9%) | 1 (0.5%) | |
| Never | 65 (39.6%) | 18 (34.0%) | 83 (38.2%) | |
| Past | 33 (20.1%) | 15 (28.3%) | 48 (22.1%) | |
| Current | 66 (40.2%) | 19 (35.8%) | 85 (39.2%) | |
| *Prescription medication use* | | | | |
| ACE inhibitor | | | | 0.196 |
| Non-user | 122 (74.4%) | 33 (62.3%) | 155 (71.4%) | |
| Past user | 20 (12.2%) | 8 (15.1%) | 28 (12.9%) | |
| Current user | 22 (13.4%) | 12 (22.6%) | 34 (15.7%) | |
| Angiotensin receptor blocker | | | | 0.708 |
| Non-user | 137 (83.5%) | 43 (81.1%) | 180 (82.9%) | |
| Past user | 3 (1.8%) | 2 (3.8%) | 5 (2.3%) | |
| Current user | 24 (14.6%) | 8 (15.1%) | 32 (14.7%) | |
| Aldosterone antagonist | | | | 0.791 |
| Non-user | 150 (91.5%) | 47 (88.7%) | 197 (90.8%) | |
| Past user | 6 (3.7%) | 3 (5.7%) | 9 (4.1%) | |
| Current user | 8 (4.9%) | 3 (5.7%) | 11 (5.1%) | |
| β-blocker | | | | 0.847 |
| Non-user | 106 (64.6%) | 32 (60.4%) | 138 (63.6%) | |
| Past user | 9 (5.5%) | 3 (5.7%) | 12 (5.5%) | |

**Table 3** Continued

| | Survivor (N=164) | Deceased (N=53) | Total (N=217) | P value |
|---|---|---|---|---|
| Current user | 49 (29.9%) | 18 (34.0%) | 67 (30.9%) | |
| Calcium channel blocker | | | | 0.233 |
| Non-user | 116 (70.7%) | 35 (66.0%) | 151 (69.6%) | |
| Past user | 14 (8.5%) | 2 (3.8%) | 16 (7.4%) | |
| Current user | 34 (20.7%) | 16 (30.2%) | 50 (23.0%) | |
| α-agonist | | | | 0.569 |
| Non-user | 163 (99.4%) | 53 (100.0%) | 216 (99.5%) | |
| Current user | 1 (0.6%) | 0 (0.0%) | 1 (0.5%) | |
| Thiazide | | | | 0.569 |
| Non-user | 163 (99.4%) | 53 (100.0%) | 216 (99.5%) | |
| Current user | 1 (0.6%) | 0 (0.0%) | 1 (0.5%) | |
| Antiplatelet | | | | 0.076 |
| Non-user | 112 (68.3%) | 28 (52.8%) | 140 (64.5%) | |
| Past user | 8 (4.9%) | 2 (3.8%) | 10 (4.6%) | |
| Current user | 44 (26.8%) | 23 (43.4%) | 67 (30.9%) | |
| Antiarrhythmic | | | | 0.07 |
| Non-user | 147 (89.6%) | 43 (81.1%) | 190 (87.6%) | |
| Past user | 6 (3.7%) | 1 (1.9%) | 7 (3.2%) | |
| Current user | 11 (6.7%) | 9 (17.0%) | 20 (9.2%) | |
| Anticoagulant | | | | 0.47 |
| Non-user | 152 (92.7%) | 47 (88.7%) | 199 (91.7%) | |
| Past user | 4 (2.4%) | 1 (1.9%) | 5 (2.3%) | |
| Current user | 8 (4.9%) | 5 (9.4%) | 13 (6.0%) | |
| Glucocorticoid | | | | 0.016 |
| Non-user | 95 (57.9%) | 19 (35.8%) | 114 (52.5%) | |
| Past user | 16 (9.8%) | 6 (11.3%) | 22 (10.1%) | |
| Current user | 53 (32.3%) | 28 (52.8%) | 81 (37.3%) | |
| β2-agonist | | | | 0.023 |
| Non-user | 131 (79.9%) | 33 (62.3%) | 164 (75.6%) | |
| Past user | 6 (3.7%) | 2 (3.8%) | 8 (3.7%) | |
| Current user | 27 (16.5%) | 18 (34.0%) | 45 (20.7%) | |
| Muscarinic antagonist | | | | 0.351 |
| Non-user | 129 (78.7%) | 39 (73.6%) | 168 (77.4%) | |
| Past user | 5 (3.0%) | 4 (7.5%) | 9 (4.1%) | |
| Current user | 30 (18.3%) | 10 (18.9%) | 40 (18.4%) | |
| NSAID | | | | 0.116 |
| Non-user | 146 (89.0%) | 43 (81.1%) | 189 (87.1%) | |
| Past user | 8 (4.9%) | 2 (3.8%) | 10 (4.6%) | |
| Current user | 10 (6.1%) | 8 (15.1%) | 18 (8.3%) | |
| Vitamin D | | | | 0.076 |
| Non-user | 109 (66.5%) | 26 (49.1%) | 135 (62.2%) | |
| Past user | 12 (7.3%) | 6 (11.3%) | 18 (8.3%) | |
| Current user | 43 (26.2%) | 21 (39.6%) | 64 (29.5%) | |
| Proton pump inhibitor | | | | 0.82 |

**Table 3** Continued

|  | Survivor | Deceased | Total |  |
| --- | --- | --- | --- | --- |
|  | (N=164) | (N=53) | (N=217) | P value |
| Non-user | 62 (37.8%) | 18 (34.0%) | 80 (36.9%) |  |
| Past user | 11 (6.7%) | 3 (5.7%) | 14 (6.5%) |  |
| Current user | 91 (55.5%) | 32 (60.4%) | 123 (56.7%) |  |
| Statin |  |  |  | 0.006 |
| Non-user | 72 (43.9%) | 11 (20.8%) | 83 (38.2%) |  |
| Past user | 12 (7.3%) | 3 (5.7%) | 15 (6.9%) |  |
| Current user | 80 (48.8%) | 39 (73.6%) | 119 (54.8%) |  |
| Immunosuppressant |  |  |  | 0.476 |
| Non-user | 156 (95.1%) | 48 (90.6%) | 204 (94.0%) |  |
| Past user | 3 (1.8%) | 2 (3.8%) | 5 (2.3%) |  |
| Current user | 5 (3.0%) | 3 (5.7%) | 8 (3.7%) |  |
| *Complications post-diagnosis* |  |  |  |  |
| Cardiovascular |  |  |  | <0.001 |
| No | 66 (40.2%) | 9 (17.0%) | 75 (34.6%) |  |
| Recurrent | 84 (51.2%) | 43 (81.1%) | 127 (58.5%) |  |
| Novel | 14 (8.5%) | 1 (1.9%) | 15 (6.9%) |  |
| Respiratory |  |  |  | 0.003 |
| No | 49 (29.9%) | 5 (9.4%) | 54 (24.9%) |  |
| Recurrent | 80 (48.8%) | 27 (50.9%) | 107 (49.3%) |  |
| Novel | 35 (21.3%) | 21 (39.6%) | 56 (25.8%) |  |
| Renal |  |  |  | <0.001 |
| No | 67 (40.9%) | 8 (15.1%) | 75 (34.6%) |  |
| Recurrent | 67 (40.9%) | 38 (71.7%) | 105 (48.4%) |  |
| Novel | 30 (18.3%) | 7 (13.2%) | 37 (17.1%) |  |
| Recurrent complications |  |  |  | <0.001 |
| None | 43 (26.2%) | 2 (3.8%) | 45 (20.7%) |  |
| 1 | 43 (26.2%) | 10 (18.9%) | 53 (24.4%) |  |
| 2 | 46 (28.0%) | 25 (47.2%) | 71 (32.7%) |  |
| 3 | 32 (19.5%) | 16 (30.2%) | 48 (22.1%) |  |
| Novel complications |  |  |  | 0.691 |
| None | 99 (60.4%) | 28 (52.8%) | 127 (58.5%) |  |
| 1 | 52 (31.7%) | 21 (39.6%) | 73 (33.6%) |  |
| 2 | 12 (7.3%) | 4 (7.5%) | 16 (7.4%) |  |
| 3 | 1 (0.6%) | 0 (0.0%) | 1 (0.5%) |  |

Values are n (%), unless otherwise specified.
*Patients with HPB cancer were not included in the non-malignant disease groups.
HPB, hepato–pancreato–biliary; NSAID, non-steroidal anti-inflammatory drug.

with higher odds of death; however, post-hoc adjustment for comorbidities returned a non-significant positive odds for these risk factors (online supplemental table 6). Stratified analyses according to HPB disease subtypes did not reveal any meaningful effect modification, principally due to small EL-HPB-COVID sample size (data not shown).

## DISCUSSION

We present, for the first time, data on a large, single-centre, multi-ethnic cohort of patients with HPB diseases, where primary, secondary and tertiary care EHRs were integrated to investigate the incidence and outcome of COVID-19, to demonstrate how key demographic characteristics and a range of comorbidities, lifestyle factors and

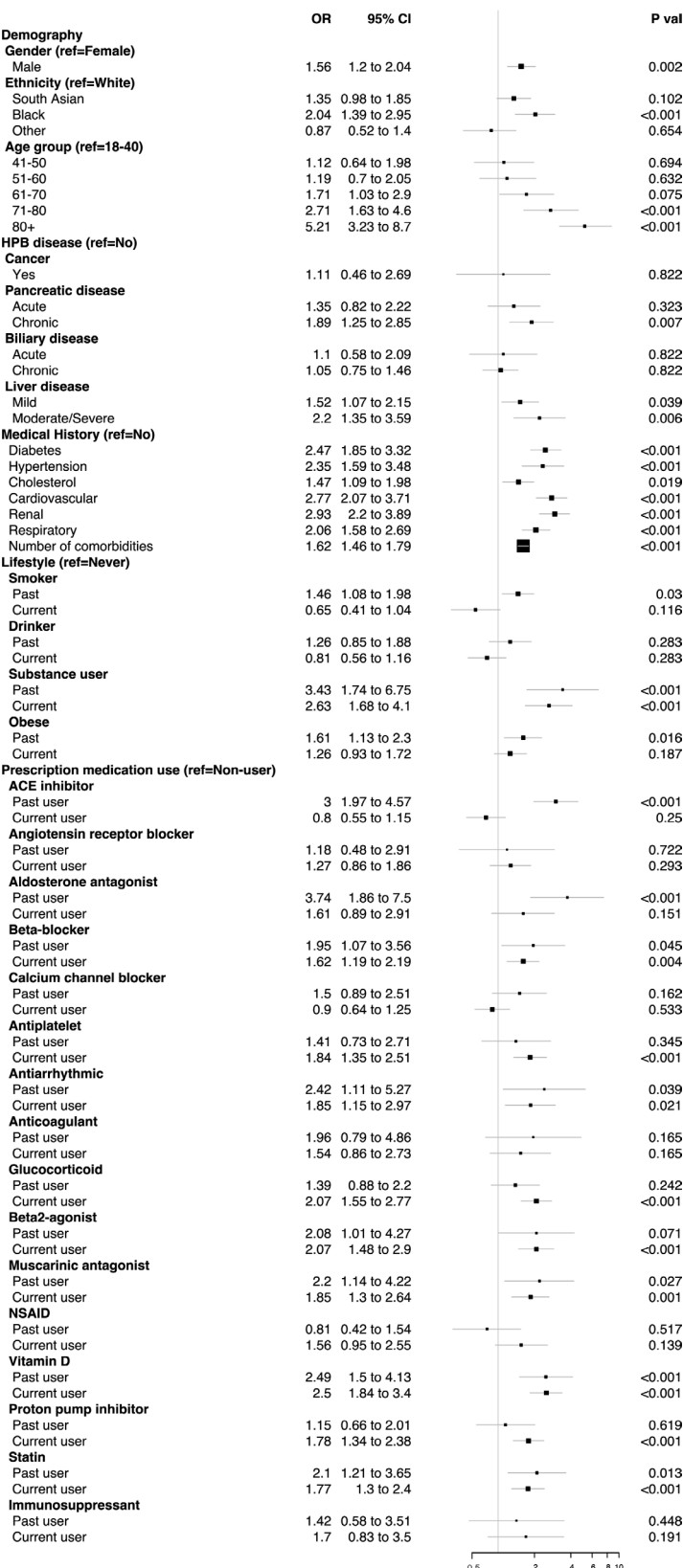

| | OR | 95% CI | P val |
|---|---|---|---|
| **Demography** | | | |
| **Gender (ref=Female)** | | | |
| Male | 1.56 | 1.2 to 2.04 | 0.002 |
| **Ethnicity (ref=White)** | | | |
| South Asian | 1.35 | 0.98 to 1.85 | 0.102 |
| Black | 2.04 | 1.39 to 2.95 | <0.001 |
| Other | 0.87 | 0.52 to 1.4 | 0.654 |
| **Age group (ref=18-40)** | | | |
| 41-50 | 1.12 | 0.64 to 1.98 | 0.694 |
| 51-60 | 1.19 | 0.7 to 2.05 | 0.632 |
| 61-70 | 1.71 | 1.03 to 2.9 | 0.075 |
| 71-80 | 2.71 | 1.63 to 4.6 | <0.001 |
| 80+ | 5.21 | 3.23 to 8.7 | <0.001 |
| **HPB disease (ref=No)** | | | |
| **Cancer** | | | |
| Yes | 1.11 | 0.46 to 2.69 | 0.822 |
| **Pancreatic disease** | | | |
| Acute | 1.35 | 0.82 to 2.22 | 0.323 |
| Chronic | 1.89 | 1.25 to 2.85 | 0.007 |
| **Biliary disease** | | | |
| Acute | 1.1 | 0.58 to 2.09 | 0.822 |
| Chronic | 1.05 | 0.75 to 1.46 | 0.822 |
| **Liver disease** | | | |
| Mild | 1.52 | 1.07 to 2.15 | 0.039 |
| Moderate/Severe | 2.2 | 1.35 to 3.59 | 0.006 |
| **Medical History (ref=No)** | | | |
| Diabetes | 2.47 | 1.85 to 3.32 | <0.001 |
| Hypertension | 2.35 | 1.59 to 3.48 | <0.001 |
| Cholesterol | 1.47 | 1.09 to 1.98 | 0.019 |
| Cardiovascular | 2.77 | 2.07 to 3.71 | <0.001 |
| Renal | 2.93 | 2.2 to 3.89 | <0.001 |
| Respiratory | 2.06 | 1.58 to 2.69 | <0.001 |
| Number of comorbidities | 1.62 | 1.46 to 1.79 | <0.001 |
| **Lifestyle (ref=Never)** | | | |
| **Smoker** | | | |
| Past | 1.46 | 1.08 to 1.98 | 0.03 |
| Current | 0.65 | 0.41 to 1.04 | 0.116 |
| **Drinker** | | | |
| Past | 1.26 | 0.85 to 1.88 | 0.283 |
| Current | 0.81 | 0.56 to 1.16 | 0.283 |
| **Substance user** | | | |
| Past | 3.43 | 1.74 to 6.75 | <0.001 |
| Current | 2.63 | 1.68 to 4.1 | <0.001 |
| **Obese** | | | |
| Past | 1.61 | 1.13 to 2.3 | 0.016 |
| Current | 1.26 | 0.93 to 1.72 | 0.187 |
| **Prescription medication use (ref=Non-user)** | | | |
| **ACE inhibitor** | | | |
| Past user | 3 | 1.97 to 4.57 | <0.001 |
| Current user | 0.8 | 0.55 to 1.15 | 0.25 |
| **Angiotensin receptor blocker** | | | |
| Past user | 1.18 | 0.48 to 2.91 | 0.722 |
| Current user | 1.27 | 0.86 to 1.86 | 0.293 |
| **Aldosterone antagonist** | | | |
| Past user | 3.74 | 1.86 to 7.5 | <0.001 |
| Current user | 1.61 | 0.89 to 2.91 | 0.151 |
| **Beta-blocker** | | | |
| Past user | 1.95 | 1.07 to 3.56 | 0.045 |
| Current user | 1.62 | 1.19 to 2.19 | 0.004 |
| **Calcium channel blocker** | | | |
| Past user | 1.5 | 0.89 to 2.51 | 0.162 |
| Current user | 0.9 | 0.64 to 1.25 | 0.533 |
| **Antiplatelet** | | | |
| Past user | 1.41 | 0.73 to 2.71 | 0.345 |
| Current user | 1.84 | 1.35 to 2.51 | <0.001 |
| **Antiarrhythmic** | | | |
| Past user | 2.42 | 1.11 to 5.27 | 0.039 |
| Current user | 1.85 | 1.15 to 2.97 | 0.021 |
| **Anticoagulant** | | | |
| Past user | 1.96 | 0.79 to 4.86 | 0.165 |
| Current user | 1.54 | 0.86 to 2.73 | 0.165 |
| **Glucocorticoid** | | | |
| Past user | 1.39 | 0.88 to 2.2 | 0.242 |
| Current user | 2.07 | 1.55 to 2.77 | <0.001 |
| **Beta2-agonist** | | | |
| Past user | 2.08 | 1.01 to 4.27 | 0.071 |
| Current user | 2.07 | 1.48 to 2.9 | <0.001 |
| **Muscarinic antagonist** | | | |
| Past user | 2.2 | 1.14 to 4.22 | 0.027 |
| Current user | 1.85 | 1.3 to 2.64 | 0.001 |
| **NSAID** | | | |
| Past user | 0.81 | 0.42 to 1.54 | 0.517 |
| Current user | 1.56 | 0.95 to 2.55 | 0.139 |
| **Vitamin D** | | | |
| Past user | 2.49 | 1.5 to 4.13 | <0.001 |
| Current user | 2.5 | 1.84 to 3.4 | <0.001 |
| **Proton pump inhibitor** | | | |
| Past user | 1.15 | 0.66 to 2.01 | 0.619 |
| Current user | 1.78 | 1.34 to 2.38 | <0.001 |
| **Statin** | | | |
| Past user | 2.1 | 1.21 to 3.65 | 0.013 |
| Current user | 1.77 | 1.3 to 2.4 | <0.001 |
| **Immunosuppressant** | | | |
| Past user | 1.42 | 0.58 to 3.51 | 0.448 |
| Current user | 1.7 | 0.83 to 3.5 | 0.191 |

**Figure 2** OR estimates of COVID-19 for patients with HPB diseases with specific demographic, comorbidity, lifestyle and medication use characteristics. OR estimates for demographic characteristics are mutually controlled for each other, that is, gender, ethnicity and age group. Estimates for HPB disease subgroups are further controlled for each other. For comorbidity, lifestyle and medication use characteristics, estimates are controlled for gender, ethnicity and dichotomous age group (under and over 60 years). HPB, hepato–pancreato–biliary; NSAID, non-steroidal anti-inflammatory drug.

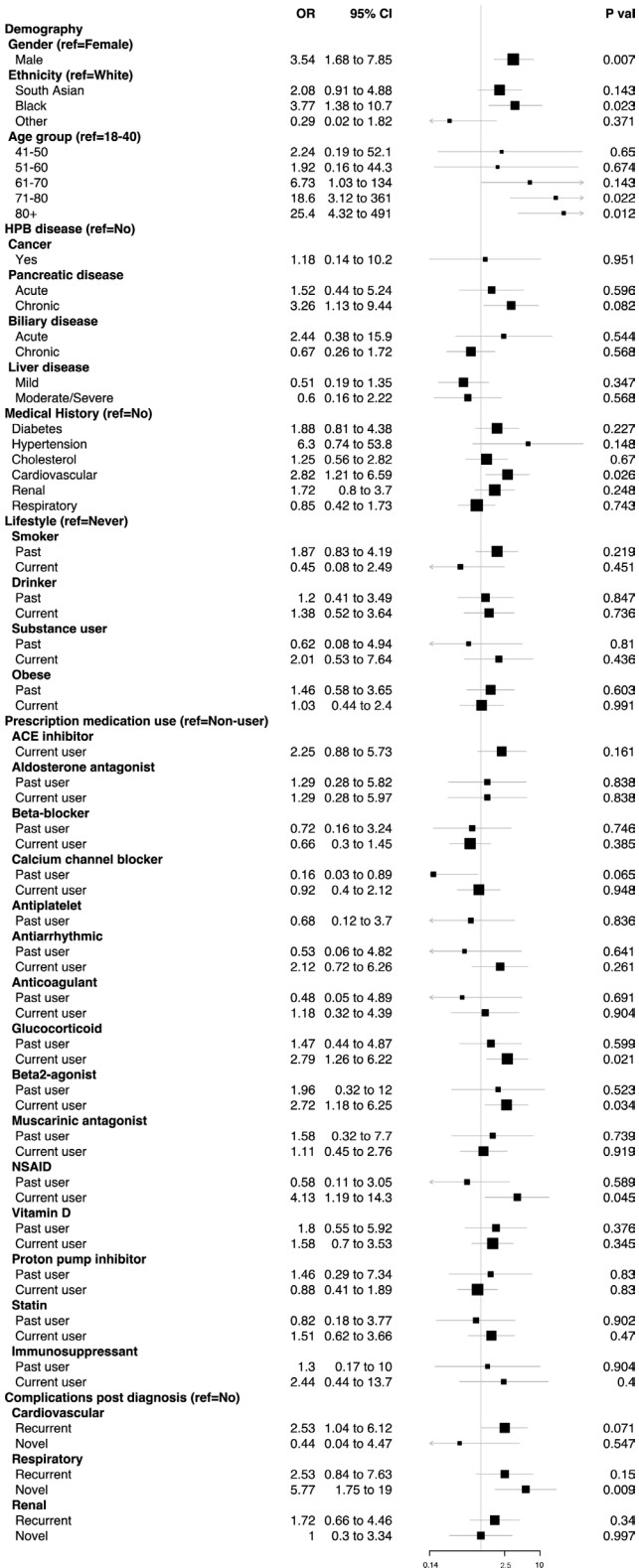

**Figure 3** OR estimates of COVID-19-related death for patients with HPB diseases with specific demographic, comorbidity, lifestyle, medication use and post-COVID-19 diagnosis complication characteristics. OR estimates for demographic characteristics are mutually controlled for each other, that is, gender, ethnicity and age group. Estimates for HPB disease subgroups are further controlled for each other. For comorbidity, lifestyle, medication use and post-diagnosis complication characteristics, estimates are controlled for gender, ethnicity and dichotomous age group (under and over 60 years). Categories with OR p>0.95 are not shown. HPB, hepato–pancreato–biliary; NSAID, non-steroidal anti-inflammatory drug.

medications are associated with SARS-CoV-2 infection and outcomes in patients with HPB diseases.

## Comparison with other studies

We noted higher odds of COVID-19 in patients with prior pancreatic and liver conditions. The higher odds associated with liver conditions are consistent with earlier findings.[6 19] Patients with moderate to severe liver conditions had higher susceptibility to SARS-CoV-2 infection than those with milder conditions, which could be due to the increase in abnormalities of immune function with severity of this disease group.[32] We speculate that reduced pancreatic function, particularly in individuals with chronic pancreatic conditions, leading to altered digestion, and therefore gut flora, may make patients more susceptible to pathogens with an enteric route of SARS-CoV-2 infection,[33 34] and also contribute to the magnitude of COVID-19 severity via modulating host-immune responses.[35] Surprisingly, the most vulnerable patients with cancer had a low COVID-19 incidence rate, which may reflect the effectiveness of public health interventions such as shielding.[36] However, at the same time, we noted a 17% death rate in this cohort (not due to COVID-19) in 6 months (online supplemental table 3), perhaps indicating the unintended, but potentially inevitable, negative sequelae of social distancing and reduced healthcare provisions for this group of patients as resources were diverted to COVID-19-affected patients.

Men had higher odds of infection and mortality than women, which is consistent with previous reports,[1 14] and could be due to a favourable genetic predisposition to the virus,[37] and/or gender differences in risk behaviours. Our study also affirms older age, particularly over 70 years, as an established risk factor for COVID-19 incidence and mortality[2 4 5]; however, this can be largely explained by the presence of multiple comorbidities in the older age groups.[38]

COVID-19 statistics have highlighted a disproportionate effect on BAME ethnic groups with an increased risk of infection and poor outcomes.[13–15] Our results confirm that people from Black community are at a higher risk of both COVID-19 infection and related mortality compared with the White ethnic group. Only a small part of the excess risk in the Black community is explained by multiple comorbidities. Therefore, further variables such as deprivation, occupational exposure and living conditions might be useful to explore as potential factors behind the apparent vulnerability of the Black population to COVID-19.

All comorbidities, such as diabetes, hypertension, high cholesterol, cardiovascular disease, kidney and respiratory disease, were independently associated with an increased risk of COVID-19, whereas the presence of cardiovascular disease contributed to an added risk of death, concurring with previously reported cohort studies.[4–6 11] Interestingly, our results highlight that for patients without underlying respiratory issues, an acute respiratory episode due to SARS-CoV-2 infection could be indicative of a worse outcome. This is in line with previous reports describing an unexpectedly lower prevalence of chronic respiratory conditions among those who had been admitted to hospital due to COVID-19[39 40]; whereas severe outcomes are often a result of respiratory complications,[41 42] such as acute respiratory distress syndrome and respiratory failure.

Smoking leads to severe health consequences, which explains the greater risk observed in our cohort of past smokers with high prevalence of respiratory and cardiovascular diseases. However, current smoking status appeared to have a protective effect in our cohort after adjusting for comorbidities, as has been observed by others, an aspect which cannot be mechanistically explained.[5 43] Carefully designed analyses are needed to explore the association and causality between smoking status (both current and past), associated comorbidities and COVID-19.

Although substance misuse leads to a plethora of cardiorespiratory and metabolic problems, its role in COVID-19 remains unexplored. To date, this is the first study providing a concrete measure of the risk of COVID-19 for substance misusers. Our initial results showing that substance misusers are at a heightened risk for COVID-19 irrespective of the comorbidities warrants a strong case for considering it as an independent risk factor for COVID-19, and may be related to high-risk behavioural patterns.[44 45]

Previous studies have found a significant relationship between obesity and an increased risk of COVID-19,[7] and subsequent hospitalisation,[46] advanced levels of treatment[15] and death.[4 6] However, our study does not suggest any particular effect of obesity on COVID-19 for patients with HPB conditions, who have a higher prevalence rate of obesity compared with the UK general population (38.8% vs 26%).[47] Our study suggests that the difference in effects for potential susceptibility to COVID-19 for patients with a history of obesity is attributed more to other prevalent factors—such as cardiovascular or chronic renal disease[47 48]—which in turn might be the consequences of obesity in these patients' lifetime.

Concerns have been raised regarding the use of various medications with respect to the risk of COVID-19 and the subsequent outcome; and, our analyses contribute to that discussion for some of the widely used prescription drugs. The higher odds observed for the history of various prescription drugs use are consistent with the management of underlying prevalent comorbidities of the study cohort: cardiovascular conditions (ACE inhibitor, β-blocker, aldosterone antagonists, antiplatelet, antiarrhythmic), cholesterol (statin), chronic respiratory diseases (glucocorticoid, β2-agonists, muscarinic antagonist), or background HPB condition (PPI). An important finding from our study is the significant risk observed for vitamin D users, supportive of the possible association between development of COVID-19 and vitamin D deficiency,[49 50] or specific medical conditions (such as kidney failure) where vitamin D prescription is prevalent. Given that BAME communities are observed

to be at a high risk of COVID-19, and there is evidence that vitamin D deficiency is particularly common in these ethnic groups,[49 50] further research on the relationship between vitamin D and COVID-19 is required, with a need to exclude confounding factors such as patients' vitamin D level. Our results also suggest that patients currently taking PPIs are more susceptible to SARS-CoV-2 infection, which concurs with a large population-based online survey conducted in the USA.[51] The use of PPIs is highly prevalent in patients with HPB diseases for the management of gastrointestinal acid-related disorders, and the finding here supports the hypothesis that current use of PPIs might influence the susceptibility to SARS-CoV-2 infection in the gastrointestinal tract through reduction of stomach acid.[51 52]

The literature is conflicted on the potential impact of antihypertensive drugs on COVID-19, particularly those that act as inhibitors to the renin–angiotensin–aldosterone system (RAAS) and upregulate ACE2 expression, suggesting these drugs may be potential risk factors for infection,[53 54] but also as having a protective effect on outcome.[55] However, recent studies found no underlying association between the use of different classes of antihypertensive drugs and the risk of developing COVID-19.[16] With a high percentage of patients with hypertension in the study cohort, our finding that a high risk of COVID-19 is associated with past intake of ACE inhibitors or aldosterone agonists is suggestive of the potential risk of switching from one class of antihypertensive drug to another. This contributes to the debate of whether discontinuation of RAAS inhibitors and considering alternative antihypertensive therapy in times of COVID-19 would be a good practice or not.[56] A marginal association of current use of ACE inhibitors with COVID-19-related death suggests that any increased risk of mortality is likely to be small and will need to be scrutinised in future as more data accumulate.

Our study also shows that recent users of antiinflammatory drugs, namely glucocorticoid and β2-agonists, had increased odds of COVID-19 and subsequent poor outcome. Controlling for comorbidities resulted in non-significant odds of infection for these patient subgroups, indicating underlying medical conditions—particularly those of respiratory system—to be responsible for the increased susceptibility. However, the observed harmful associations between these drugs and COVID-19-related death could not be explained by a simplified binary representation of underlying six common health conditions. Glucocorticoid drugs, for instance, are used to treat many other inflammatory conditions, notably inflammatory bowel disease (IBD), whereas HPB diseases constitute some of the most common extraintestinal manifestations of IBD. It has been shown that use of corticosteroids is associated with adverse COVID-19 outcomes among patients with IBD.[17] Had it been possible to successfully control for differences in respiratory disease severity or other medical comorbidities, we speculate to see different and possibly non-significant odds of death in these patient subgroups.

## Strengths and limitations of the study

A key strength of our study is that we have systematically identified the effect, or the lack of it, of individual demographic and clinical factors on the infection and mortality of COVID-19 in a cohort of over 15 000 patients, robustly corrected for potential confounders in their evaluation. Our large population is highly representative of patients with HPB diseases from diverse ethnic groups, which contributes to the generalisability of our findings. Another strength is our use of linked EHRs, harmonised for variations in coding that exist between different EHR systems. We ascertained patient demographics, lifestyle, comorbidities and medications by linking hospital records with pseudo-anonymised longitudinal primary care records, which substantially enrich the data that are recorded on hospital visits.

Retrospective EHR-based COVID-19 studies often suffer from incomplete or missing data on patient characteristics, including key variables such as BMI, ethnicity, smoking or pre-existing comorbidities.[4 57] The missing data are particularly applicable to otherwise healthy patients with COVID-19 with low use of healthcare services in the past. However, our patient cohort had already been treated or managed at BHNT hospitals at least once, and often referred through primary care, which led to near-complete data for this study, an added advantage of this study. For instance, ethnicity, a common demographic feature, is missing only for 2.8% of cases in our cohort while the rate is significantly higher in other studies (up to 20% of cases).[4 57] The only variable with missing data frequency over 20% in our study is substance misuse behaviour (50.4%). This is a unique lifestyle risk variable which is not yet explored—understandably due to a lack of recorded data as people often do not disclose this information to their clinicians,[58] unless manifested in physical or mental disorders. Yet, the substance misuse history of over 7600 patients included in this study provides a good indication of the impact of COVID-19 on this understudied group.

Our study also has some important limitations. One limitation is the risk of residual confounding or confounding by indication due to unmeasured or simplified binary representation of potential confounding variables. For example, the observed association between vitamin D users and risk of COVID-19 may be different if participants' vitamin D level/deficiency status had been taken into consideration. Similarly, the observed association between COVID-19-related death and recent use of glucocorticoid or β2-agonists may reduce or get amplified if respiratory disease severity or other indications for corticosteroid use were considered.

Another critical limitation is associated with the confirmation of East London residency for the study cohort. Patients' addresses (current or past) are not collected under the umbrella study, which considers patients with HPB conditions (with the exception of cancer) treated or managed at BHNT hospitals as East London residents during the time of their care. The Royal London Hospital

hosts one of the largest HPB centres in England, and supports patients with suspected or confirmed HPB cancer from nearby geographical areas. As the umbrella study cohort is historical, we acknowledged the probability of people moving away from East London in the meantime. In absence of a patient's current address to confirm their residency at the outset of COVID-19 pandemic in the UK, we relied on an indirect measure to infer residency. We used a strict 6-month window preceding the study to identify a patient's interaction with East London GPs or BHNT hospitals. Thus, we believe that any supposed reduction in the cohort size due to unaccounted change of residency within that window should have affected the COVID-19 and non-COVID-19 group in equal proportion, and hence unlikely to alter the findings we report here.

Due to the rarity of the outcome (SARS-CoV-2 infection) in the full HPB cohort, the effects reported in the study could be influenced by the smaller cohort size of COVID-19 cases. We recognise that larger sample sizes of patients with COVID-19 are needed to fully understand the effect of SARS-CoV-2 in patients with HPB conditions. Our results are the first step towards this and require validation in similar national and international cohorts.

## CONCLUSIONS

We believe that the findings from this single-centre study, focusing on patients with a particular medical condition and in an ethnically diverse area, highlight some considerations that could guide clinical care while we await an effective antiviral strategy for COVID-19. The current findings reinforce our understanding of some of the important risk factors for SARS-CoV-2 infection but with regard to pre-existing HPB conditions, and allow stratification for risk, thereby providing a tool for policymakers to divert prevention as well as treatment to a clearly identified vulnerable population.

**Acknowledgements** AZMDU is supported by Health Data Research UK (HDR-UK) to conduct the umbrella study EL-PaC-Epidem, which is funded by the UK Medical Research Council. We gratefully acknowledge support provided by the Pancreatic Cancer Research Fund (PCRF), for conducting public–patient engagement activity and facilitating ethical approval for EL-PaC-Epidem. We thank Dr Charles Gutteridge, Chief Clinical Information Officer at Barts Health NHS Trust, for his help with the collection of secondary and tertiary care data. We thank Dr Kambiz Boomla, Dr John Robson, Professor Carol Dezateux, and members of the Discovery East London Programme Board and developers at Learning Health Solutions for their support in facilitating collection of primary care patient records. Finally, we acknowledge the contribution to the research made by several members of the PCRF Tissue Bank team, Bioinformatics Unit, and clinical research fellows at Barts Cancer Institute through insightful medical and scientific discussion.

**Contributors** AZMDU designed the study, and was responsible for undertaking and completing data collection, processing and analysis. HK and CC oversaw the conduct and management of the study. AZMDU, LS, HK and CC contributed to the selection of study variables and interpretation post-analysis. AZMDU wrote the first drafts of the report and all the authors made critical revisions.

**Funding** The study is conducted under an umbrella study, focusing on the epidemiology of pancreatic and other hepatobiliary cancers in East London (EL-PaC-Epidem), funded by the Medical Research Council UK (Ref: MR/S003835/1) as a UKRI/Rutherford Fellowship to the corresponding author. No additional funding has been received for this study.

**Competing interests** None declared.

**Patient consent for publication** Not required.

**Ethics approval** All data used for this study were collected and processed under the EL-PaC-Epidem Study at Barts Health NHS Trust. The study was approved by the East of England-Essex Research Ethics Committee (19/EE/0163; 17 May 2019) and supported by the NHS Confidentiality Advisory Group for collecting and processing confidential patient information without consent (19/CAG/0219; 17 January 2020).

**Provenance and peer review** Not commissioned; externally peer reviewed.

**Data availability statement** All statistical data relevant to the study are included in the article or uploaded as supplemental information. Only the corresponding author had full access to all the participants' data in the study. The authors confirm that researchers seeking the completely anonymised final analysis dataset for this work can submit a data request to the corresponding author.

**ORCID iDs**
Abu Z M Dayem Ullah http://orcid.org/0000-0002-2567-4648
Hemant M Kocher http://orcid.org/0000-0001-6771-1905

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
