## [Reviewer comments · BMJ Open]

ARTICLE DETAILS

TITLE (PROVISIONAL)	COVID-19 in patients with hepatobiliary and pancreatic diseases: A single-centre cross-sectional study in East London
AUTHORS	Dayem Ullah, Abu; Sivapalan, Lavanya; Kocher, Hemant; Chelala, Claude

VERSION 1 – REVIEW

REVIEWER	Luca Gianotti School of Medicine and Surgery, University of Milano-Bicocca, Italy
REVIEW RETURNED	12-Oct-2020

GENERAL COMMENTS	COVID-19 in patients with hepatobiliary and pancreatic diseases: A single-centre cohort study in East London By Abu Z M Dayem Ullah et al. This represents a timely manuscript and potentially addressing a important topic. This reviewer rises a major concern about the reliability and interpretation of the results: the authors aim to explore risk factors associated with COVID-19 susceptibility and survival in patients with pre-existing hepato-pancreato-biliary (HPB) conditions. HPB conditions included 11 different benign pancreatic disease coding, 48 for liver conditions and 19 for biliary diseases. The case mix and the severity of the diseases are very wide going from (as an example)focal nodular hyperplasia of liver to acute and subacute liver necrosis. This is an important confounding variable for both COVID-19 susceptibility and survival.
---

REVIEWER	John Gubatan, MD Stanford University School of Medicine, USA
REVIEW RETURNED	06-Jan-2021

GENERAL COMMENTS	In this large, multicenter, population-based retrospective cohort study from the United Kingdom, Ullah et al sought to determine clinical risk factors associated with COVID-19 susceptibility and mortality among patients with hepato-pancreato-biliary (HPB) diseases. This study addresses an important and timely topic and may help clinicians risk stratify susceptible patient populations. The study is overall well-designed and reported, has a large sample size, and employed appropriate statistical analyses. I have the following critiques and recommendations:
---

	1. Results: The authors should report the background prevalence of COVID-19 among the general population in their cohort. How does this compare with rates of COVID-19 among HPB patients in their registry? 2. The authors demonstrate that patients with liver and pancreatic disease are at increased risk of COVID-19 compared to the other HPB patients. However, liver and pancreatic disease is a broad classification that consists of very heterogeneous diagnoses each with different risks and pathogenesis. The authors should perform subgroup analyses to better refine and compare risk of COVID-19 susceptibility and mortality among the liver and pancreatic disease groups. Consider breaking down risks in liver group (by cirrhosis, hepatitis, hepatocellular carcinoma, etc) and likewise pancreatic disease (acute pancreatitis, chronic pancreatitis, pancreatic cancer, etc). 3. Discussion: The discussion should provide more explanations of why patients with liver and pancreatic disease may be at higher risk for COVID-19. 4. Results/Discussion: It is interesting that current vitamin D use was associated with increased risk for COVID-19. Vitamin D deficiency has been associated with risk of severe COVID-19 (Jain, A., Chaurasia, R., Sengar, N.S., Singh, M., Mahor, S. and Narain, S., 2020. Analysis of vitamin D level among asymptomatic and critically ill COVID-19 patients and its correlation with inflammatory markers. Scientific reports, 10(1), pp.1-8.). Thus, vitamin D level is a likely confounder. Vitamin D levels should be adjusted in their multivariate analysis for COVID-19 susceptibility and mortality. It is unclear if current vitamin D use was associated with increased risk of COVID-19 due to low levels among vitamin D users. 5. Discussion: There were several medications (glucocorticoid steroids, statins, B-blockers, antiplatelets, muscarinic antagonists, etc) that were associated risk of COVID-19 among HPB patients. These should be discussed further. For example, how does the risk of glucocorticoid with COVID-19 compare with other patients with gastrointestinal disease (e.g. inflammatory bowel disease)?
--	--

REVIEWER	Anna Stachel NYU Langone Health, USA
REVIEW RETURNED	27-Jan-2021

GENERAL COMMENTS	It is good to see a comprehensive analysis of Covid-19 in a sub-population. My overarching concerns/clarifications needed include: 1) is study design/analysis truly longitudinal? even though data were collected longitudinally, I think it was analyzed as a cross-sectional 2.) why is HPB diagnosis used in all the adjustments? I thought the study population is HPB patients? 3.) Can the definition of death be clarified further? Is there a bias to detect more deaths that happened in February 2020 compared to June 2020? For example, someone with covid in Feb has 5 months to be counted as a death compared to 1 month if they were diagnosed in June. (assuming the count happened in July.). Also, how quickly does the death registry update information? Specific comments attached.
--

	- The reviewer provided a marked copy with additional comments. Please contact the publisher for full details.
--	--

VERSION 1 – AUTHOR RESPONSE

Reviewer: 1

=====

>> This represents a timely manuscript and potentially addressing a important topic.

We thank the reviewer for the positive comment.

>> This reviewer rises a major concern about the reliability and interpretation of the results: the authors aim to explore risk factors associated with COVID-19 susceptibility and survival in patients with pre-existing hepato-pancreato-biliary (HPB) conditions. HPB conditions included 11 different benign pancreatic disease coding, 48 for liver conditions and 19 for biliary diseases. The case mix and the severity of the diseases are very wide going from (as an example) focal nodular hyperplasia of liver to acute and subacute liver necrosis. This is an important confounding variable for both COVID-19 susceptibility and survival.

We appreciate the reviewer’s concern. The same concern was echoed by Reviewer 2. We acknowledge the limitation of adopting a broad and binary classification (presence or absence) of the pancreatic, liver and biliary diseases for exploring risks of COVID-19 susceptibility and mortality. This was due to the relatively small number of COVID-19 cases under each disease category. Following reviewers’ suggestions, we have re-classified our non-malignant HPB patient cohort into different subgroups. Non-malignant liver diseases were divided into mild and moderate to severe subgroups, extending the definition from CDMF Charlson Comorbidity Index; whereas non-malignant pancreas or biliary diseases were divided into acute and chronic disease subgroups (Methods section; supplemental table 1). Within each disease category, patients with a history of both acute (or mild) and chronic (more severe) conditions were assigned to the chronic (or more severe) subgroup. All patients with a diagnosed HPB cancer were grouped under a separate Cancer category.

The risk estimates for each disease subgroup was presented in reference to the absence of the particular disease (e.g., chronic pancreatic condition vs no pancreatic disease). We have also explored potential association between HPB diseases and COVID-19 susceptibility/mortality risk factors in stratified analyses according to the disease subgroups (supplemental table 6). The stratified analysis was not conducted for Cancer category due to very small number of COVID-19 cases (N=6).

Reviewer: 2

=====

>> In this large, multicenter, population-based retrospective cohort study from the United Kingdom, Ullah et al sought to determine clinical risk factors associated with COVID-19 susceptibility and mortality among patients with hepato-pancreato-biliary (HPB) diseases. This study addresses an important and timely topic and may help clinicians risk stratify susceptible patient populations. The study is overall well-designed and reported, has a large sample size, and employed appropriate statistical analyses.

We thank the reviewer for the positive comment.

>> I have the following critiques and recommendations:

>> 1. Results: The authors should report the background prevalence of COVID-19 among the general population in their cohort. How does this compare with rates of COVID-19 among HPB patients in their registry?

This has been added to the Results section. During the study period, the prevalence of COVID-19 in the background East London population was 41 per 10 000 adult population, whereas the rate was more than three times higher (145 per 10 000 adult people) in our HPB disease cohort. The age-standardised rate could not be calculated as the population age profile could not be obtained for the three boroughs of East London.

>> 2. The authors demonstrate that patients with liver and pancreatic disease are at increased risk of COVID-19 compared to the other HPB patients. However, liver and pancreatic disease is a broad classification that consists of very heterogeneous diagnoses each with different risks and pathogenesis. The authors should perform subgroup analyses to better refine and compare risk of COVID-19 susceptibility and mortality among the liver and pancreatic disease groups. Consider breaking down risks in liver group (by cirrhosis, hepatitis, hepatocellular carcinoma, etc) and likewise pancreatic disease (acute pancreatitis, chronic pancreatitis, pancreatic cancer, etc).

This has now been addressed. Please see our response to Reviewer 1 above.

>> 3. Discussion: The discussion should provide more explanations of why patients with liver and pancreatic disease may be at higher risk for COVID-19.

We have now added the explanation below to the Discussion section:

“We noted a higher odds of COVID-19 in patients with prior pancreatic and liver conditions. The higher odds associated with liver conditions is consistent with earlier findings.⁶ 19 Patients with moderate to severe liver conditions had higher susceptibility to SARS-CoV2 infection than those with milder conditions, which could be due to the increase in abnormalities of immune function with severity of this disease group³². We speculate that reduced pancreatic function, particularly in individuals with chronic pancreatic conditions, leading to altered digestion, and therefore gut flora, may make patients more susceptible to pathogens with an enteric route of SARS-CoV2 infection³³ 34, and also contribute to the magnitude of COVID-19 severity via modulating host immune responses³⁵.”

>> 4. Results/Discussion: It is interesting that current vitamin D use was associated with increased risk for COVID-19. Vitamin D deficiency has been associated with risk of severe COVID-19 (Jain, A., Chaurasia, R., Sengar, N.S., Singh, M., Mahor, S. and Narain, S., 2020. Analysis of vitamin D level among asymptomatic and critically ill COVID-19 patients and its correlation with inflammatory markers. Scientific reports, 10(1), pp.1-8.). Thus, vitamin D level is a likely confounder. Vitamin D levels should be adjusted in their multivariate analysis for COVID-19 susceptibility and mortality. It is unclear if current vitamin D use was associated with increased risk of COVID-19 due to low levels among vitamin D users.

We have updated the Discussion section to address this comment. Briefly, we acknowledge that the observed risk for vitamin D users is “supportive of the possible association between development of COVID-19 and vitamin D deficiency, 49 50 or specific medical conditions (such as kidney failure) where Vitamin D prescription is prevalent.” We also speculate that there could be a three-way association among BAME communities, vitamin D level and risk of COVID-19, as mentioned in the Discussions section: “Given that BAME communities are observed to be at a high risk of COVID-19,

and there is evidence that vitamin D deficiency is particularly common in these ethnic groups,⁴⁹ further research on the relationship between vitamin D and COVID-19 is required, with a need to exclude confounding factors such as patients' vitamin D level."

Unfortunately, we did not have access to the data on the vitamin D level of participants, therefore it was not possible to include patients' vitamin D level as confounding variables for the observed association. For the possible confounding effect of renal condition where Vitamin D prescription is prevalent, we have shown in the Results section that the "slightly reduced yet significantly high odds remained after controlling for comorbidities, with renal (for Vitamin D users) diseases being the principal source of the reduced estimate (supplemental table 4)". A cursory look at our data shows that Vitamin D use is particularly high among South Asian participants compared to White population (21% vs 17.5%), it was not significant to warrant changes in the risk estimate.

Finally, we have acknowledged the limitation of this unmeasured confounding in the Discussion section (Strengths and Limitations of the Study subsection): "the observed association between Vitamin D users and risk of COVID-19 may be different if participants' Vitamin D level/deficiency status had been taken into consideration."

>> 5. Discussion: There were several medications (glucocorticoid steroids, statins, B-blockers, antiplatelets, muscarinic antagonists, etc) that were associated risk of COVID-19 among HPB patients. These should be discussed further. For example, how does the risk of glucocorticoid with COVID-19 compare with other patients with gastrointestinal disease (e.g. inflammatory bowel disease)?

Both Results and Discussion sections have been updated to address this comment. The "higher odds observed for the history of various prescription drugs use are consistent with the management of underlying prevalent comorbidities of the study cohort: cardiovascular conditions (ACE inhibitor, β -blocker, aldosterone antagonists, antiplatelet, antiarrhythmic), cholesterol (statin), chronic respiratory diseases (glucocorticoid, β 2-agonists, muscarinic antagonist), or background HPB condition (proton pump inhibitor)" is explained by the non-significant or reduced risk estimate when controlled for the respective medical conditions.

Our updated results in light of the changes in sample size (as mentioned before) also show that "recent users of anti-inflammatory drugs, namely glucocorticoid and β 2-agonists, had increased odds for COVID-19 and subsequent poor outcome". We found that controlling for comorbidities resulted in non-significant odds of infection for these patient subgroups, indicating underlying medical conditions - particularly weak respiratory system - to be responsible for the increased susceptibility. However, as we explained in the Discussion section, "the observed harmful associations between these drugs and COVID-19-related death could not be explained by a simplified binary representation of underlying six common health conditions. Glucocorticoid drugs, for instance, are used to treat many other inflammatory conditions, notably inflammatory bowel disease (IBD), whereas HPB diseases constitute some of the most common extraintestinal manifestations of IBD. It has been shown that use of corticosteroids is associated with adverse COVID-19 outcomes among patients with IBD.¹⁷ Had it been possible to successfully control for differences in respiratory disease severity or other medical comorbidities, we speculate to see different and possibly non-significant odds of death in these patient subgroups."

Again, we have acknowledged the limitation of this unmeasured confounding in the Discussion section (Strengths and Limitations of the Study subsection).

Reviewer: 3

=====

>> It is good to see a comprehensive analysis of Covid-19 in a sub-population.

We thank the reviewer for the positive comment.

>> My overarching concerns/clarifications needed include:

>> 1) is study design/analysis truly longitudinal? even though data were collected longitudinally, I think it was analyzed as a cross-sectional

We agree with the reviewer. Even though the study participants represent a retrospective cohort (i.e., patients diagnosed with HPB condition between 1 April 2008 and 6 March 2020), the subsequent data analysis for COVID-19 susceptibility and mortality reflects a cross-sectional design. This has now been acknowledged in the Title, Abstract and Methods section of the manuscript. The effect size for each variable under investigation is now evaluated and presented with odds ratios (ORs), instead of risk ratios (RRs).

>> 2.) why is HPB diagnosis used in all the adjustments? I thought the study population is HPB patients?

The study cohort represents HPB patients from different disease category: HPB Cancers, and non-malignant pancreatic diseases, liver diseases, biliary diseases. Adjusting for HPB diagnoses according to the categorisation was an attempt to minimise the effect on risk estimate (if any) due to the variability in underlying HPB diseases distribution. We agree with the reviewer that this possibly contributed to over-adjustment. Hence, the revised analyses do not include HPB diagnosis variables in the multivariable regression models.

HPB disease variables are still included in the regression models when estimating risks for HPB disease subgroups. A participant with non-malignant HPB diagnoses for multiple organs can be represented in multiple HPB subgroups. Therefore, the effect estimation for individual HPB disease variables is further mutually controlled for other HPB diseases, along with the demographic variables.

>> 3.) Can the definition of death be clarified further? Is there a bias to detect more deaths that happened in February 2020 compared to June 2020? For example, someone with covid in Feb has 5 months to be counted as a death compared to 1 month if they were diagnosed in June. (assuming the count happened in July.). Also, how quickly does the death registry update information?

We thank the reviewer for critical observation. Indeed, the association between COVID-19 mortality and explored risk factors could be biased by a more generous approach of defining all patients who eventually died after COVID-19 diagnosis within the study period (12 February to 12 June 2020) as COVID-19-related death. We have now adopted the standard definition of COVID-19 related death and refined our COVID-19 deceased group with those who died within 28 days of their first COVID-19 diagnosis. To ensure consistency, a 60-day window was being used in the UK prior to 12 August 2020 (covering the study period) to define COVID-19 related death. To ensure clarity, COVID-19 patients who survived beyond 60 days of index date are considered as survivors in the study; Those who died between 29 and 60 days of diagnosis were excluded from the analysis (figure 1; CONSORT flow diagram).

In order to ensure all COVID-19 related death - particularly those who were diagnosed near the end of the study period (12 June 2020) – were correctly accounted for in our mortality analysis, we have collected updated mortality data on 12 October 2020. We collate mortality data from three different

sources: hospital records, GP records and central National Health Service (NHS) Spine system via Personal Demographics Service (PDS). It can take between 24 hours and seven days for a mortality event to be reflected in one of these systems. We believe a four-month window after the study end period is broad enough to capture all relevant mortality events.

>> 4) A series of minor issues were raised as annotated comments in the original manuscript.

We have addressed those additional comments:

- Six common medical comorbidities (diabetes, hypertension, high cholesterol, cardiovascular disease, chronic respiratory disease, renal disease) were explored in this study for possible association with SARS-CoV-2 infection and related death. Although this was presented in Table 1, we have now clarified it further in the text (Study design and population subsection). While many other risk factors (e.g., dementia) are definitely worth exploring for COVID-19 research, we concentrated our focus on the above-mentioned six.
- We conducted an automated sub-string search within semi-structured text entries (discharge summaries, past medical history and a lifestyle questionnaire collected during the pre-operative assessment, and presenting symptoms from scheduled or unscheduled hospital visits) to identify relevant events for some variables (clarified in Procedures subsection).
- Ethnicity, rather than race, is the widely accepted term in the UK National Health Service (NHS) to denote various ethnic groups.
- The analyses now included comparison of mortality rate between COVID-19 and non-COVID-19 groups (supplementary table 3; Results section; Population characteristics subsection).
- Further details have now been included to address the association between gut flora and COVID-19 disease severity with updated references (Discussion section).
- The observed association between COVID-19 and past obesity is addressed in the Discussion section. We have shown that this could rather be attributed to other prevalent factors – such as cardiovascular or chronic renal disease– which in turn might be the consequences of obesity in these patients' lifetime.

VERSION 2 – REVIEW

REVIEWER	Gubatan, John Stanford University School of Medicine, Gastroenterology and Hepatology
REVIEW RETURNED	08-Mar-2021
GENERAL COMMENTS	The authors have suitably responded to the reviewer's critiques and have significantly improved the manuscript.